# Optimising acute stroke care organisation: a simulation study to assess the potential to increase intravenous thrombolysis rates and patient gains

Maarten M H Lahr  ,[1] Durk-Jouke van der Zee,[2] Gert-Jan Luijckx,[3] Erik Buskens[1,3]

¹Department of Epidemiology, University of Groningen, University Medical Centre Groningen, Groningen, The Netherlands
²Department of Operations, Faculty of Economics and Business, University of Groningen, Groningen, The Netherlands
³Department of Neurology, University of Groningen, University Medical Centre Groningen, Groningen, The Netherlands

**Correspondence to**
Dr Maarten M H Lahr;
m.m.h.lahr@umcg.nl

## ABSTRACT

**Objectives** To assess potential increases in intravenous thrombolysis (IVT) rates given particular interventions in the stroke care pathway.

**Design** Simulation modelling was used to compare the performance of the current pathway, best practices based on literature review and an optimised model.

**Setting** Four hospitals located in the North of the Netherlands, as part of a centralised organisational model.

**Participants** Ischaemic stroke patients prospectively ascertained from February to August 2010.

**Intervention** The interventions investigated included efforts aimed at patient response and mode of referral, prehospital triage and intrahospital delays.

**Primary and secondary outcome measures** The primary outcome measure was thrombolysis utilisation. Secondary measures were onset-treatment time (OTT) and the proportion of patients with excellent functional outcome (modified Rankin scale (mRS) 0–1) at 90 days.

**Results** Of 280 patients with ischaemic stroke, 125 (44.6%) arrived at the hospital within 4.5 hours, and 61 (21.8%) received IVT. The largest improvements in IVT treatment rates, OTT and the proportion of patients with mRS scores of 0–1 can be expected when patient response is limited to 15 min (IVT rate +5.8%; OTT −6 min; excellent mRS scores +0.2%), door-to-needle time to 20 min (IVT rate +4.8%; OTT −28 min; excellent mRS scores+3.2%) and 911 calls are increased to 60% (IVT rate +2.9%; OTT −2 min; excellent mRS scores+0.2%). The combined implementation of all potential best practices could increase IVT rates by 19.7% and reduce OTT by 56 min.

**Conclusions** Improving IVT rates to well above 30% appears possible if all known best practices are implemented.

## INTRODUCTION

Intravenous thrombolysis (IVT) is an effective therapy for acute ischaemic stroke up to 4.5 hours after the onset of symptoms.[1 2] This therapy is substantially underused, however, with 8%–10% of all stroke patients worldwide

### Strengths and limitations of the study

► The simulation modelling study included a comprehensive collection of patient level data from both the prehospital and intrahospital acute stroke pathway.

► A simulation-based approach, as presented in this paper, can be instrumental in facilitating a broad overview of the set-up and performance of stroke pathways.

► Effects of capacity constraints on patients' waiting for care services are not explicitly modelled given their high priorities, allowing them to queue jump.

► Costs items associated with the proposed interventions could not be collected and controlled for.

► Estimations of time intervals used for model building might have changed over time.

currently receiving IVT.[3 4] In contrast, treatment rates of up to 35% have been achieved in optimised settings.[5 6] The organisation of stroke care is an important factor in realising timely hospital arrival and treatment.[7 8] The centralisation of care at designated stroke centres has been demonstrated to increase the proportion of patients arriving at the hospital in time for acute treatment.[9–11] Given the substantial decrease in the benefit of treatment with increasing time delays, further efforts to expedite hospital arrival and subsequent treatment remain of crucial importance.[12 13]

Various studies have investigated factors associated with efficiency in each part of the acute stroke pathway. Although it has been generally established that delay on the part of patients and/or bystanders is a primary factor in delaying hospital arrival,[14] interventions aimed at improving optimal response by calling 911 immediately have exhibited varying success, and many lack sustained

implementation[15] (as is the case in other domains of medicine as well).[16] Ambulance transportation to hospitals that offer acute treatment is associated with shorter onset-to-door times, reductions in intrahospital delays and increases in treatment rates.[15 17 18] The provision of acute treatment by a mobile stroke unit (MSU) before hospital arrival has been identified as a promising method for reducing time to IVT.[19] Another widely studied topic concerns reducing the time between hospital arrival and treatment (door-to-needle (DTN) time), with reported DTN times as low as 20 min.[20]

The aforementioned studies have demonstrated clear benefits in terms of time saved. They nevertheless lack a broader overview of pathway set-up and its performance. Instead of addressing the solution space as a whole, they target isolated elements of the stroke pathway. The lack of a broader overview is due in part to the predominant use of randomised controlled trials (RCT) as the main research vehicle. Given the effort involved in their set-up, RCT studies focus predominantly on separate and singular elements of pathway performance. They may therefore be less suitable for investigating complex care systems (eg, acute stroke treatment). In particular, timely hospital arrival and the treatment of acute stroke patients relies on a series of intertwined activities concerning patient diagnostics and transportation.

One potential alternative methodology is simulation modelling. Proceeding from a detailed description of both prehospital and intrahospital time delays and diagnostics, an accurate representation of pathway performance can be developed in silico, including the validation of IVT rates, time to treatment, patient outcomes (as measured by the modified Rankin Scale (mRS)) and other clinically relevant outcome measures.[21 22] This approach would allow the examination of all time delays and diagnostic steps, thereby providing clinicians and policymakers with an accurate overview of obstacles currently existing within care pathways. In addition, scenarios for hypothetical approaches to improvement based on clinical guidelines, literature observations and/or expert opinion can be modelled and studied for their cumulative effects on relevant clinical outcome measures.[23 24]

The aims of this simulation-modelling study were (1) to estimate the cumulative potential for improving IVT utilisation by implementing best practices on the organisation of the stroke pathway and, subsequently, (2) to explore areas in which further improvement is needed in order to achieve a fully optimised setting, in addition to identifying obstacles to such optimisation.

## METHODS

This article is based on a 6-month, prospective, multicentre study performed in a centralised organisational stroke-care setting in the north of the Netherlands from February through July 2010.[9] Patient-level data were collected on time delays and diagnostics, thereby providing detailed insight into patient flow and potential

| Table 1 Descriptive statistics of activity durations and diagnostics | |
|---|---|
| Number of patients | 280 |
| Age in years (SD) | 70 (14) |
| Male (%) | 156 (56) |
| Patient responsiveness | |
| Time from symptom onset to call for help, valid cases (%) | 152 (54) |
| Median, min (IQR) | 41 (5–130) |
| Mode of referral (%) | |
| General practitioner | 129 (46) |
| 911 | 84 (30) |
| Self-referral | 60 (21) |
| In-hospital patients | 7 (3) |
| Pathway set-up | |
| Transported by EMS (%) | 213 (76) |
| Median response time, min (IQR) | 9 (7–12) |
| Median on scene time, min (IQR) | 20 (15–25) |
| Median transportation time, min (IQR) | 17 (9–22) |
| Median time from hospital arrival to neurological examination, min (range) | 2 (0–15) |
| Median time from hospital arrival to CT examination min (IQR) | 12 (6–15) |
| Median time from hospital arrival to laboratory examination, min (IQR) | 32 (27–37) |
| Median door to IVT time, min (IQR) | 35 (25–45) |

EMS, emergency medical services; IVT, intravenous thrombolysis.

obstacles in both the prehospital and intrahospital pathways (table 1). A schematic overview of patient flow and the steps included in the analyses is presented in figure 1.

### Setting and participants

The centralised organisational setting consisted of four hospitals in the northern region of the Netherlands, with IVT being administered only in the University Medical Center Groningen (UMCG). Together with the other three community hospitals, general practitioner (GP) offices and emergency medical services (EMS), arrangements were made to transport presumed stroke victims immediately to the UMCG, thus bypassing community hospitals that might have been located closer to the patient's location. The international protocol for IVT (adjusted European Cooperative Acute Stroke Study (ECASS) III[25] and the regional protocol for prehospital management were followed. The region addressed in this study comprises approximately 580 000 inhabitants, with a population density of 250 inhabitants/km². The study population consisted of ischaemic stroke patients admitted to all four hospitals between February and August 2010. Case ascertainment was confirmed by the

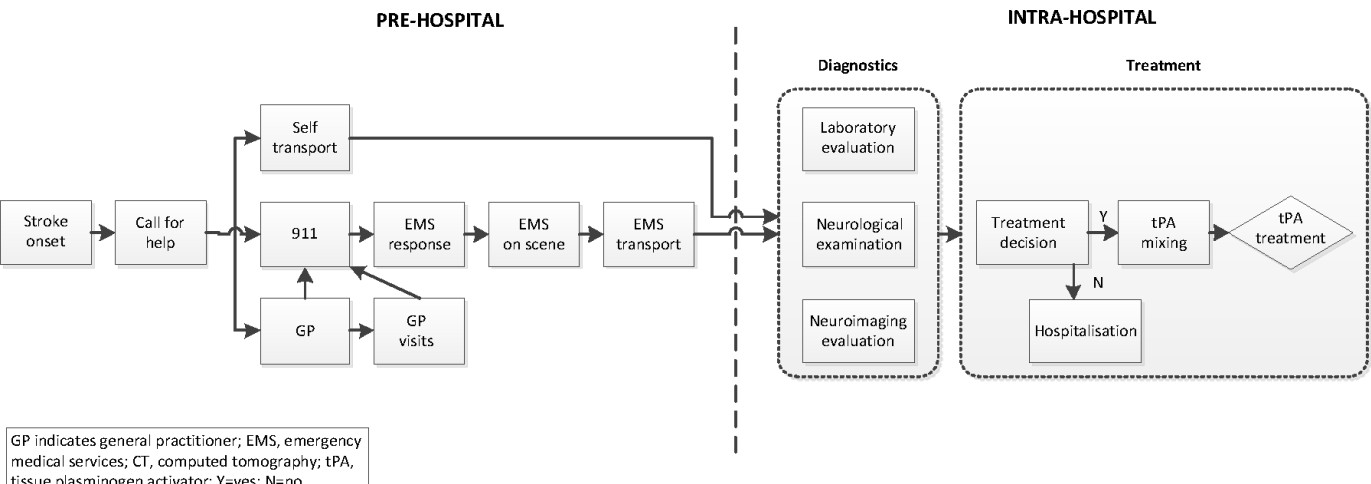

PRE-HOSPITAL

INTRA-HOSPITAL

Diagnostics

Treatment

GP indicates general practitioner; EMS, emergency medical services; CT, computed tomography; tPA, tissue plasminogen activator; Y=yes; N=no

**Figure 1** Acute stroke pathway: description of activities. EMS, emergency medical services; GP, general practitioner; N, no; tPA, tissue plasminogen activator; Y, yes.

final hospital discharge diagnosis of ischaemic stroke, thereby excluding stroke mimics.

The data collected included information on time delays and diagnostics along the entire stroke chain. Stroke care pathways can be described in several distinct phases: hyperacute (emergency), acute and rehabilitation. Within this study we focused on the hyperacute phase, ranging from symptom onset until acute treatment with IVT. Time delays included the time from symptom onset to call for help, delay at first response (GP or EMS services), EMS transport times and intrahospital diagnostics, which included time from hospital arrival to neurological examination, CT scan, laboratory testing and IVT.

### Simulation model

The current study was performed using a previously validated simulation model.[24] More detailed information on simulation modelling methodology, input parameters, model and model data used may be found in a online supplementary file. This model was populated with patient-level data from a previous observational study.[9] The model was validated by comparing IVT treatment percentages and onset-treatment time (OTT) to those reported in the observational study. The next step consisted of developing scenarios in which alternative pathway set-ups, associated time delays and diagnostics were imputed based on literature observations, clinical guidelines and expert opinion. Using the simulation model, hypothetical patients were passed through the system, estimating the impact of intervening at various points in the acute stroke pathway. Interventions were modelled by changing the underlying statistical distributions to redistribute patients across time delays and diagnostics.[26]

### Scenarios

We used simulation modelling to investigate the effects of changing pathway set-ups, based on three models: (1) a baseline model of acute stroke care; (2) a model reflecting best practices, based on a review of the current

literature, clinical guidelines and expert opinion; and (3) an optimised model. Interventions were selected according to the obstacles identified in the current centralised organisational model. Obstacles within the study setting included delayed emergency response by patients following symptom onset, mode of referral (GP or 911), time spent on the scene by ambulance personnel and intrahospital delays. The simulation model was used to perform hypothetical interventions in the pathway to calculate clinically relevant outcomes.[24] The outcomes were compared with the baseline performance of the current system to estimate the potential for improvement, to optimise system performance, and to identify obstacles that have yet to be overcome.

### Baseline

The baseline model describes the performance of the centralised organisational model for acute stroke care, as described in the previous observational study.[9] A description of time delays and diagnostics along each step of the pathway is provided in table 1.

### Best practice

The scenarios and input parameters that were investigated are described in table 2.

*Patient responsiveness*: We estimated the relative impact of reducing the time between stroke onset and call for help by patients, their families and/or bystanders by adjusting the distribution of their response times by a factor equal to the quotient of the respective median response times reported for best practices (15–30 min)[27 28] and the baseline scenario (41 min).

*Mode of referral*: We modelled a scenario in which patients, their families and/or bystanders predominantly (60%) chose 911 as the mode of entry.[29 30]

*Time spent on the scene by ambulance personnel*: In this scenario, we modelled the time spent on the scene by ambulance personnel following a 911 call by imposing an upper boundary on on-scene delay. Based on the guidelines of the American Stroke Association, the time spent on the

**Table 2** Overview of scenarios and input parameters

| Factor | Baseline | Scenarios | | Study | Input parameters |
|---|---|---|---|---|---|
| **Patient responsiveness** | | | | | |
| 1. Patient delay | 41 min* | A. Reduced to 30 min (*best practice*)<br>B. Reduced to 15 min (*best practice*)<br>C. Reduced to 0 min (*optimised*) | | Salisbury *et al*[27]<br>Carroll *et al*[28] | Time from stroke onset to call for help |
| 2. Mode of referral 911 | 30% | A. Transport by EMS increased to 60% (*best practice*)<br>B. Transport by EMS increased to 100% (*optimised*) | | Barsan *et al*[29], Chen *et al*[30] | Choice of route, choice of first responder |
| 3. Combined best practices patient responsiveness | | Combines scenarios 1B and 2A | | | See factors 1 and 2 |
| 4. Optimised patient responsiveness | | Combines scenarios 1C and 2B | | | See factors 1 and 2 |
| Pathway set-up | | | | | |
| 5. Response time first responders | 9 min* | Response time first responders reduced to 0 min (*optimised*) | | | Delay first responder |
| 6. On scene time ambulance personnel | 20 min* | A. Reduced to 15 min (*best practice*)<br>B. Reduced to 10 min (*best practice*)<br>C. Reduced to 0 min (*optimised*) | | Jauch *et al*[37], Acker *et al*[38],<br>Atkins *et al*[36] | EMS—time spent on scene |
| 7. Prehospital IVT by MSU transportation time | 17 min* | A. Transportation time reduced to 0 min (*best practice*)<br>B. All ambulance delays reduced to 0 min (*optimised*) | | Fassbender *et al*[19] | EMS—response time (7A,7B),<br>EMS—transport Time (7B) |
| 8. Door to IVT times | 35 min* | A. Reduced to 30 min (*best practice*)<br>B. Reduced to 25 min (*best practice*)<br>C. Reduced to 20 min (*best practice*)<br>D. Reduced to 0 min (*optimised*) | | Zinkstok *et al*[33],<br>Meretoja *et al*[34],<br>Meretoja *et al*[20] | Time to neurological consultation, time to neuroimaging examination, time to laboratory examination, decision making, IVT mixing |
| 9. Combined best practices pathway set-up | | Combines scenarios 6B, 7A and 8C | | | See factors 6, 7 and 8 |
| 10. Optimised pathway set-up | | Combines scenarios 5, 6C, 7B and 8D | | | See factors 5, 6, 7 and 8 |
| **Patient responsiveness and pathway set-up** | | | | | |
| 11. Combined best practices patient responsiveness and pathway set-up | | Combines scenarios 3 and 9 | | | See factors 3 and 9 |
| 12. Optimised patient responsiveness and pathway set-up | | Combines scenarios 4 and 10 | | | See factors 4 and 10 |

*Median.
EMS, emergency medical services; IVT, intravenous thrombolysis; MSU, mobile stroke unit.

scene should be no more than 15 min.[31] We also modelled the implementation of a 'scoop and run protocol', which includes prompt transport (≤10 min) with initial management efforts, while postponing elaborate triage.

*Prehospital treatment by MSU:* In this scenario, we modelled prehospital IVT provided by a MSU by adjusting only the transport time of the ambulance. Assuming that the MSU would be stationed at the hospital, we considered only

the transportation time from the hospital to the patient scene.

*Intrahospital processes:* Intrahospital delays comprise all activities performed between arrival at the hospital until the start of IVT (DTN time; see figure 1). Based on the guidelines of the American Stroke Association, DTN time should be no more than 60 min.[32] Evidence from clinical practice suggests that DTN times within a range of 20–30 min are attainable.[20 33 34]

### Combined best-practice scenarios

Three scenarios were performed in which best practices were combined for patient responsiveness and pathway set-up. Patient responsiveness was modelled by combining patient response with the mode of referral following stroke onset. Pathway set-up was modelled by combining prehospital and intrahospital best practices. A third scenario combines best practices for patient responsiveness and pathway set-up.

### Optimised scenarios

Additional scenarios were defined to interpret findings on the effects of the implementation of best practices (or combinations thereof) by generating upper boundaries to pathway performance, thereby building on unrealistically optimistic assumptions. For the optimised scenarios, we extrapolated best practices by setting the parameter for associated time delays to 0 or by setting the diagnostic quality parameter to 100%.

### Outcome measures

The primary outcome measure was the proportion of patients treated with IVT. Secondary outcome measures were the total process time (onset-to-treatment time), IVT within various time intervals (0–90, 91–180 and 181–270 min), the proportion of patients with favourable outcomes at 90 days (mRS 0–1) and additional healthy life days (calculated using OTT estimates).[12 35]

### Analysis

For each of the scenarios described above, we calculated new hypothetical IVT treatment rates and secondary outcome measures, based on the number of patients arriving in time for acute treatment at the hospital (ie, within 4 hours after symptom onset), the number treated with IVT and the time to treatment. $\chi^2$ tests were used to compare categorical variables.

### Ethics approval and patient consent

Informed consent was obtained from all subjects participating in the prospective study[9] and extended for the current simulation study.

### Patient and public involvement

Patients and public were not involved in the design of this study. For this modelling study non-identifiable patient data was used. Study results will be disseminated through poster presentations and publication in peer-reviewed journals.

## RESULTS

This study reflects the experiences of 280 ischaemic stroke patients referred to the UMCG and three community hospitals, as part of a centralised organisational model. Baseline and demographic characteristics are described in table 1. In all, 125 (44.6%) patients arrived at the hospital within 4.5 hours, 61 (21.8%) received IVT, and the median OTT was 127 min. Of the patients receiving IVT, 17.0% were treated within 90 min of symptom onset. Patient delay, intrahospital delays and mode of referral (GP or 911) were identified as the greatest obstacles to receiving IVT (table 1).

### Simulation experiments

The results of the simulation experiments are presented in table 3.

*Patient responsiveness:* If patients had contacted emergency services sooner (ie, within 30 and 15 min), up to 27.6% (CI 26.7% to 28.4%) of the total population would have been treated with IVT (table 3, Scenarios 1A and 1B), and the OTT would have been reduced to 122 min (CI 121 to 124).[27 28]

Assuming a patient delay of 0 min (table 3, Scenario 1C), 64.0% of the total population would have been treated with IVT, and the OTT would have been reduced to 92 min (CI 91 to 92).

*Mode of referral:* Assuming 60% of all patients contacting 911 immediately following stroke onset (table 3, Scenario 2A) increased the IVT rate to 24.6% (CI 23.8% to 25.5%) and reduced the OTT to 127 min (CI 125 to 128).[29]

If all patients (100%) had called 911 (table 3, Scenario 2B), the IVT rates would have increased further to 28.4% (CI 27.5% to 29.3%), and the OTT would have been reduced to 124 min (CI 123 to 126).

*Time spent on the scene by ambulance personnel:* Shortening the time spent on the scene to 15 and 10 min (table 3, Scenarios 6A and 6B),[36] increased the IVT treatment rate up to 23.3% (CI 22.4% to 24.1%) and reduced the OTT to 121 min (CI 120 to 123).[37 38]

Reducing time on the scene to 0 min resulted in a projected IVT rate of 24.7% (CI 23.8% to 25.5%) (table 3, Scenario 6C) and a decrease in the OTT to 114 min (CI 112 to 116).

*Prehospital treatment by MSU:* In this set-up, 23.2% (CI 22.4% to 24.0%) of patients would have been treated with IVT (table 3, Scenario 7A) and OTT would have been reduced to 121 min (CI 120 to 123).

The elimination of both the response time and transportation time of the MSU (table 3, Scenario 7B) resulted in a projected 25.6% (CI 24.7% to 26.4%) of patients that could have received IVT and a reduction of the OTT to 109 min (CI 107 to 110).

*Intrahospital processes:* If the DTN time had been shortened to a maximum of 30 and 20 min (table 3, Scenarios 8A–C), up to 26.6% (CI 25.7% to 27.4%) of the total population would have been treated with IVT, and the OTT would have been reduced to 101 min (CI 99 to 103).

**Table 3** Re-configuration centralised model: results simulation experiments

| Scenario | IVT rate (95% CI) | OTT min (95% CI) | IVT 0–1.5 hours (%) | IVT 1.5–3.0 hours (%) | IVT 3.0–4.5 hours (%) | mRS 0–1 (%)* | Extra healthy days† |
|---|---|---|---|---|---|---|---|
| **0. Current practice** | 21.8% (20.9% to 22.6%) | 129 (127 to 130) | 17.0 | 70.9 | 12.1 | 12.5 | |
| **Patient responsiveness** | | | | | | | |
| **1. Patient delay** | | | | | | | |
| A. Reduced to 30 min | 23.7% (22.9% to 24.6%) | 127 (125 to 129) | 16.9 | 72.2 | 10.9 | 12.6 | 2.9 |
| B. Reduced to 15 min | 27.6% (26.7% to 28.4%)‡ | 122 (121 to 124) | 16.5 | 76.7 | 6.8 | 12.7 | 11.3 |
| C. Reduced to 0 min | 64.0% (63.0% to 64.9%)‡ | 92 (91 to 92) | 45.7 | 54.1 | 0.2 | 16.2 | 66.9 |
| **2. Mode of referral 911** | | | | | | | |
| A. Increased to 60% | 24.6% (23.8% to 25.5%) | 127 (125 to 128) | 18.8 | 70.0 | 11.2 | 12.8 | 3.9 |
| B. Increased to 100% | 28.4% (27.5% to 29.3%)‡ | 124 (123 to 126) | 18.3 | 71.4 | 10.3 | 12.7 | 7.8 |
| **3. Combined best practices patient responsiveness (1B+2A)** | 31.0% (30.1% to 31.9%)‡ | 120 (118 to 121) | 19.2 | 74.4 | 6.4 | 13.0 | 16.3 |
| **4. Optimised patient responsiveness (1C+2B)** | 64.3% (63.3% to 65.2%)‡ | 98 (97 to 98) | 36.2 | 63.7 | 0.1 | 15.1 | 55.7 |
| **Pathway set-up** | | | | | | | |
| **5. Response time first responders reduced to 0 min** | 23.3% (22.4% to 24.1%) | 121 (119 to 122) | 21.8 | 68.9 | 9.3 | 13.2 | 14.4 |
| **6. On scene time ambulance personnel** | | | | | | | |
| A. Reduced to 15 min | 22.8% (21.9% to 23.6%) | 124 (122 to 126) | 19.4 | 69.9 | 10.7 | 12.8 | 8.6 |
| B. Reduced to 10 min | 23.3% (22.4% to 24.1%) | 121 (120 to 123) | 21.7 | 68.5 | 9.8 | 13.1 | 13.4 |
| C. Reduced to 0 min | 24.7% (23.8% to 25.5%) | 114 (112 to 116) | 31.4 | 60.4 | 8.2 | 14.3 | 26.3 |
| **7. Prehospital IVT by MSU** | | | | | | | |
| A. Transportation time reduced to 0 min | 23.2% (22.4% to 24.0%) | 121 (120 to 123) | 22.8 | 67.4 | 9.8 | 13.2 | 13.1 |
| B. All ambulance delays reduced to 0 min | 25.6% (24.7% to 26.4%) | 109 (107 to 110) | 38.6 | 54.3 | 7.0 | 15.1 | 35.8 |
| **8. Door to IVT times** | | | | | | | |
| A. Reduced to 30 min | 25.2% (24.3% to 26.0%) | 110 (108 to 111) | 34.3 | 57.7 | 8.0 | 14.6 | 34.4 |
| B. Reduced to 25 min | 25.9% (25.0% to 26.8%) | 106 (104 to 107) | 39.3 | 53.2 | 7.5 | 15.2 | 41.5 |
| C. Reduced to 20 min | 26.6% (25.7% to 27.4%) | 101 (99 to 103) | 44.2 | 48.8 | 7.0 | 15.7 | 49.8 |
| D. Reduced to 0 min | 29.8% (28.9% to 30.7%)‡ | 83 (81 to 85) | 62.9 | 32.1 | 5.0 | 17.9 | 81.9 |

Continued

**Table 3** Continued

| | IVT rate (95% CI) | OTT min (95% CI) | IVT 0–1.5hours (%) | IVT 1.5–3.0hours (%) | IVT 3.0–4.5 hours (%) | mRS 0–1 (%)* | Extra healthy days† |
|---|---|---|---|---|---|---|---|
| 9. Combined best practices pathway set-up (6B+7A+8C) | 30.8% (29.9% to 31.7%)‡ | 80 (79 to 82) | 68.9 | 26.3 | 4.8 | 18.6 | 87.3 |
| 10. Optimised pathway set-up (5+6C+7B+8D) | 38.5% (37.6% to 39.5%)‡ | 39 (37 to 40) | 86.0 | 11.3 | 2.7 | 20.6 | 161.7 |
| Patient responsiveness and pathway set-up | | | | | | | |
| 11. Combined best practices patient responsiveness and pathway set-up (3+9) | 41.5% (40.5% to 42.4%)‡ | 73 (72 to 74) | 77.4 | 20.5 | 2.2 | 19.6 | 99.9 |
| 12. Optimised patient responsiveness and pathway set-up (4+10) | 97.7% (97.4% to 98.0%)‡ | 0 (0 to 0) | 100.0 | 0.0 | 0.0 | 22.2 | 231.6 |

*Indicates the proportion of patients with good outcome (mRS 0–1) ascribed to treatment with thrombolysis.[33]
†Indicated the number of additional days in healthy life by minute reduction in OTT.[1]
‡Sig. <0.05.
IVT, intravenous thrombolysis; mRS, modified Rankin scale; MSU, mobile stroke unit; OTT, onset-treatment time.

If the DTN had been reduced to 0 min (table 3, Scenario 8D), 29.8% (CI 28.9% to 30.7%) of all patients would have been treated with IVT, and the OTT would have been reduced to 83 min (CI 81 to 85).

### Combined best practice scenarios

Combining best practices for patient responsiveness and pathway set-up (table 3, scenario's 3, 9, 11) resulted in up to 41.5% (CI 40.5% to 42.4%) of all patients being treated with IVT and reduced the OTT to 73 min (CI 72 to 74).

### Optimised scenarios

Assuming optimised patient responsiveness (ie, all patients calling 911 immediately following stroke onset; table 3, Scenario 4) resulted in 64.3% of the total population (CI 63.3% to 65.2%) being treated with IVT and reduced the OTT to 98 min (CI 97 to 98). The optimisation of pathway set-up (table 3, Scenario 10) resulted in 38.5% (CI 37.6% to 39.5%) of all patients receiving IVT and reduced the OTT to 39 min (CI 37 to 40). The combination of all optimised scenarios (table 3, Scenario 12) resulted in a cumulative total of 97.7% (CI 97.4% to 98.0%) of all patients being treated with IVT and reduced the OTT to 0 min.

### DISCUSSION

This study demonstrates that IVT treatment rates above 30% would be possible if best practices were to be implemented within our setting. We modelled several scenarios to generate insight into the potential for quality improvements in our acute stroke chain of care. Although improvements in patient responsiveness would yield the largest potential gains within our pathway, even modest changes in this regard are likely to be challenging and costly to achieve.[16] In contrast, improvements in other areas (eg, intrahospital delays and time spent by ambulance personnel at the patient's location) might be easier to achieve and would still lead to clinically relevant increases in IVT rates. As indicated in previous studies, even small reductions in time to treatment with IVT are associated with considerable increases in the length of healthy life, and they may require only relatively simple organisational changes involving minimal effort at little or no cost.[12]

The results of our study may be useful as a guide for prioritising interventions along the acute stroke pathway and for estimating their potential impact on the effectiveness of the pathway. A simulation-based approach, as presented in this paper, can be instrumental in facilitating a broad overview of the set-up and performance of stroke pathways. This could provide clinicians and policymakers with speedy answers—at little effort or cost—concerning how new or widely advocated practices could be used to improve their pathways, thus allowing them to direct investments to the interventions that matter most. It thus has the potential to replace RCT studies or serve as a precursor to a focussed RCT, which could be scoped as the net result of a simulation approach.

In our study, we observed the greatest effects on IVT rates, time to treatment and patient outcomes after improving the responsiveness of patients and/or bystanders by reducing the time from symptom onset to the call for help, thereby expediting intrahospital care services and by increasing the number of 911 calls made by patients or bystanders. In contrast, a scenario in which prehospital transportation delays were reduced by the implementation of a MSU resulted in only moderate effects. Combining all of the best-practice scenarios resulted in a maximum of 41.5% of patients being treated with IVT. This is substantially higher than current benchmark figures on clinical practice, which suggest a maximum IVT rate of around 35%.[6 39]

Proceeding from a scenario in which best practices have been implemented, remaining challenges include realising further decreases in time delays in both the prehospital and intrahospital phases. The feasibility of such initiatives in clinical practice might be limited in the short term, however, given the current lack of evidence concerning solutions for further expediting care and logistics services at reasonable costs. For example, the goal of reducing time spent on the scene by ambulance personnel to less than 10 min or reducing DTN time to less than 20 min would probably be unrealistic, given the need to handle and observe the patient, to complete diagnostic tests and to interpret findings. Although further improvements in the proportion of patients calling 911 directly following symptom onset could potentially result in further increases in IVT rates, they would also necessitate large-scale and repetitive publicity campaigns comparable to those launched to raise public awareness on stroke symptoms and how to act.

The organisational model for acute stroke care delivery is currently receiving a great deal of attention in the Netherlands, as well as beyond.[40 41] The emergence of endovascular treatment (EVT) for patients facing large-vessel occlusions has opened up a whole new dimension in terms of acute stroke pathway set-up and patient logistics. Following IVT, eligible patients must now undergo additional diagnostic evaluation (eg, CT angiography and perfusion CT), followed by such EVT treatment modalities as groin punctures and initial attempt at clot retrieval with the device up to the angioseal following successful recanalisation. In addition, within the current 'drip-and-ship' treatment paradigm, eligible patients may initially be admitted to community hospitals before being transferred to comprehensive stroke centres with EVT capacity, thereby further increasing the number of logistical steps. Given the time-sensitive nature of acute stroke interventions, this extension of the pathway necessitates the re-organisation of acute stroke care within regions and settings. In this respect, simulation modelling could facilitate insight into the complex interplay of separate elements of the pathway. Currently positioned as a follow-up treatment by current guidelines, availability of EVT does not change the need for optimising utilisation of IVT, nor does it impact the acute stroke pathway set-up

for IVT. Moreover, the subgroup of patients eligible for EVT is relatively small, around 7% of all stroke patients.[42]

Our study is subject to several limitations. First, our simulation models did not consider the response of GPs when contacted as first responders. Although this has been signalled as an issue for delays in hospital arrival for patients,[43] no studies on best practices were identified in the literature. Second, because the costs and cost-effectiveness associated with pathway improvements in our setting were not estimated in our model, it was not possible to control for them. Third, the results of our findings might not be generalisable to other settings, due to the unique position of the UMCG, which serves as a stroke centre in a centralised organisational model deployed within its region. As noted in a previous publication, however, the generic modelling approach adopted in this study could be extended to include a description of a decentralised organisational model, which receives IVT candidates within its own catchment areas.[23] Also, because patients were enrolled in the observational study back in 2010 these results will not fully reflect current practice. However, review of internal databases show that IVT treatment percentages have remained largely stable over the last years, fluctuating around 25% with a DTN time of around 35 min. EMS response times have remained constant over the years.[44] In addition, pathway set-up of acute stroke patients receiving IVT remained similar over the years. Finally, model assumptions excluded the possibility for capacity constraints influencing patient waiting times, as patients and/or transport queuing seldom occurs due to the high prioritisation that potential acute stroke patients receive throughout the pathway. However, we acknowledge that such constraints might occur in other stroke care systems. Future activities should be aimed at extending the simulation-based approach to include the drip-and-ship model currently employed in acute stroke treatment (eg, EVT).

## CONCLUSIONS

The results of this study indicate that the cumulative effects of implementing best practices on the organisation of stroke care would clearly exceed current benchmarks for treatment rates. Remaining obstacles might be difficult to overcome given the limited availability of solution to further expedite care and logistical services at tolerable costs. A broader overview facilitated by simulation is suggested as instrumental in supporting decision-makers and clinicians in their efforts to evaluate the set-up and performance of acute stroke pathways.

**Contributors** MMHL and D-JZ designed the study, performed the experiments and analysed the data. MMHL drafted the manuscript, D-JZ, G-JL and EB critically revised the manuscript for intellectual content and approved the final version of the manuscript for publication.

**Funding** The authors have not declared a specific grant for this research from any funding agency in the public, commercial or not-for-profit sectors.

**Competing interests** None declared.

**Patient consent for publication** Not required.

**Ethics approval** No additional approval was necessary because for this simulation study an anonymised dataset was used.

**Provenance and peer review** Not commissioned; externally peer reviewed.

**Data availability statement** Data are available upon reasonable request.

**Open access** This is an open access article distributed in accordance with the Creative Commons Attribution 4.0 Unported (CC BY 4.0) license, which permits others to copy, redistribute, remix, transform and build upon this work for any purpose, provided the original work is properly cited, a link to the licence is given, and indication of whether changes were made. See: https://creativecommons.org/licenses/by/4.0/.

**ORCID iD**
Maarten M H Lahr http://orcid.org/0000-0001-7265-2612

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
