## [Reviewer comments · BMJ Open]

ARTICLE DETAILS

TITLE (PROVISIONAL)	Optimizing acute stroke care organization: a simulation study to assess the potential to increase intravenous thrombolysis rates and patient gains
AUTHORS	Lahr, Maarten; van der Zee, Durk-Jouke; Lujckx, Gert-Jan; Buskens, Erik

VERSION 1 – REVIEW

REVIEWER	Prof Christos Vasilakis University of Bath, UK
REVIEW RETURNED	17-Jul-2019

GENERAL COMMENTS	1. Title: Spell out IVT 2. Abstract, Participants: this is rather old data, does it have an impact on the quality of values for the model's input parameters or the context of interventions to be tested? 3. There is no mention of thrombectomy and how the introduction of this new clinical intervention may affect the policies tested out in this study. 4. Methods: I am afraid the model is under-reported and a lot more detail is needed here. The list is long but indicatively, we need to know the modelling approach (e.g. discrete-event or time-driven), the number of replications per scenario, whether the simulation was a steady-state or transient, if steady state then the warm-up period used, the time horizon of the simulation, the statistical methods used to calculate output measures and to compare the output from different scenarios, the simulation package or programming language the model was implemented in. I am not certain an exemplar paper is needed as the authors have already cited a number of them. In any case, a decent further example can be found here: https://www.ncbi.nlm.nih.gov/pubmed/21344201 If word count limit is an issue, then consider adding a supplementary file (similar to paper already cited: 10.1371/journal.pone.0079049). The simulation code could also be included as part of this effort. 5. Methods, page 6, line 27: it is mentioned that the model study was performed using a previously validated model. A reference is given in the next sentence citing the study that collected and analysed the data. However, no reference is given to the paper that reported the validation of the simulation model. 6. Methods: continuing on the previous theme, and in addition to any references provided in response to my previous comment, the input parameters, the distribution shape and values as used in the baseline model and the scenarios, as well as their sources should be included in one (or more) comprehensive tables. Table 1 is potentially a starting point, but this refers to the data collected
---

	prospectively in the four centres, rather than what went into the simulation model. 7. Figure 1: is this a high-level depiction of the pathway or the actual process flow implemented as part of the simulation model? In any case, as part of describing the simulation model we need to see the specific process map that acted as a blueprint for the model to be developed. 8. Table 2: the under-reporting of the simulation model and its constituent elements makes it rather difficult to fully understand this part of the study. That said, it appears that many 'delays' were used as input rather than output parameters from the simulation (e.g. for validation purposes). In reality, 'delays' are typically the result of the mismatch between demand and supply, or of the way a care process is organised (or both). If feeding delays in to the model as an input parameter, then how would this parameter change if for example, we were to change the way the care process is organised or vary arrivals rates etc.? Isn't the whole idea of modelling to be able to assess the impact of changes on delays/waiting times (as well as other output parameters?) How valid is the model, if the delay/waiting time distributions are exogenous rather than generated by the model itself? 9. Figure 2. The chart is not very clear and this is not only because of how small it is on paper. In any case, I am reserving judgement on the results section of the paper until my comments above are addressed/rebutted satisfactorily.
--	---

REVIEWER	Dalius Jatuzis Vilnius University Faculty of Medicine, Institute of Clinical Medicine, Clinics of Neurology and Neurosurgery, Lithuania
REVIEW RETURNED	14-Aug-2019

GENERAL COMMENTS	Thank you for possibility to review your manuscript. It is really very important research for optimising the delivery of acute stroke care. My comments:  1. I would suggest briefly describing the statistical methods used in the work (even if they were fairly simple in this work). 2. The introduction could be slightly shortened, focusing on the main idea and task of the work. 3. Did you not need the ethics approval for this work, in your opinion? If so, I suggest you mention it in the text.
--

VERSION 1 – AUTHOR RESPONSE

Reply to reviewers' comments

Prof Christos Vasilakis (Reviewer 1)

Comment 1: Title: Spell out IVT

Response of the authors:

We agree with the comment of the reviewer. We have therefore spelled out IVT in the title as: “intravenous thrombolysis” [Unblinded title page, line 2].

Comment 2: Abstract, Participants: this is rather old data, does it have an impact on the quality of values for the model’s input parameters or the context of interventions to be tested?

Response of the authors:

We agree with the reviewer that the data we used as input might be perceived as seemingly old. However, the current organization of care, i.e., the flow of patients in the system that was studied has not changed. It is the same centralized organisational model for acute stroke care. No changes in the set-up of the pathway for acute stroke patients receiving intravenous thrombolysis have been implemented over the years. As such the context has remained similar.

The clinical data used as input for generating model outcomes represent current clinical practice, as already in 2010, an optimised system with a relatively short door-to-needle time of around 35 minutes was realized. In addition, we know that infrastructure and ambulance services are stable and thus patient arrival times at the hospital and emergency medical services transportation times have remained constant over the years. No large interventions in either public educational campaigns or at the emergency medical services have taken place.

To clarify the above mentioned points, we added two sentences to the Discussion section. It now reads: “..with a DTN time of around 35 minutes. EMS response times have remained constant over the years.⁴⁴” [Blinded manuscript, Discussion, page 15, line 14-15].

Comment 3: There is no mention of thrombectomy and how the introduction of this new clinical intervention may affect the policies tested out in this study.

Response of the authors:

We thank the reviewer for raising this point. Therefore we added a sentence to the Discussion section clarifying the issue raised. It now reads: “Currently positioned as a follow-up treatment by current guidelines, availability of EVT does not change the need for optimizing utilization of IVT, nor does it impact the acute stroke pathway set-up for IVT. Moreover, the subgroup of patients eligible for EVT is relatively small, around 7% of all stroke patients.⁴²” [Blinded manuscript, Discussion, page 14, line 21-24].

Comment 4: I am afraid the model is under-reported and a lot more detail is needed here. The list is long but indicatively, we need to know the modelling approach (e.g. discrete-event or time-driven), the number of replications per scenario, whether the simulation was a steady-state or transient, if steady state then the warm-up period used, the time horizon of the simulation, the statistical methods used to calculate output measures and to compare the output from different scenarios, the simulation package or programming language the model was implemented in.

I am not certain an exemplar paper is needed as the authors have already cited a number of them. In any case, a decent further example can be found here:

<https://eur03.safelinks.protection.outlook.com/?url=https%3A%2F%2Fwww.ncbi.nlm.nih.gov%2Fpubmed%2F21344201&data=02%7C01%7Cm.m.h.lahr%40umcg.nl%7C74171cc85c5f4e551e5608d737669764%7C335122f9d4f44d67a2fccd6dc20dde70%7C0%7C0%7C637038786077355389&sdata=IRVddfrWyMUa971YMuuyS2U9IAz1Ww51FSHAPIMbIZA%3D&reserved=0>

If word count limit is an issue, then consider adding a supplementary file (similar to paper already cited: 10.1371/journal.pone.0079049).

The simulation code could also be included as part of this effort.

Response of the authors:

We agree with the reviewer that the simulation modeling methodology could have been described in more detail, and are grateful for pointing this out.

Due to the constraints in terms of word count limit (and in line with the reviewer's suggestion) we propose adding a supplementary file. This file provides the details requested by the reviewer [Supplementary file_rebuttal]. We also added a sentence to the Methods section explaining where the reader can find additional information. It now reads: "More detailed information on simulation modeling methodology, input parameters, model and model data used may be found in a supplementary file." [Methods, page 6, line 8-10]. We also included a sentence on the statistical analysis used to compare scenario results. It now reads: "Chi-square tests were used to compare categorical variables" [Methods, page 9, line 15-16]. We also added the results of the statistical analyses to Table 3 for IVT rate [Table 3. Main text].

We adopted Plant Simulation as a tool for modelling and simulating the acute stroke pathway. Using this tool model coding essentially boils down to parametrization of pre-defined building blocks, mainly concerning basic process elements and distributions specifying activity durations and diagnostics. Hence, the structure of the model is rather simple, and there is little code to interpret. We are prepared to make the coded model available upon reasonable request. In our view, the model is well explained by the process model (see Figure S1. Supplementary file_rebuttal) together with the overview of distributions provided in the supplementary file (Table S1. Supplementary file_rebuttal) and the overview of scenarios and input parameters (Table S2. Supplementary file_rebuttal).

Comment 5: Methods, page 6, line 27: it is mentioned that the model study was performed using a previously validated model. A reference is given in the next sentence citing the study that collected and analysed the data. However, no reference is given to the paper that reported the validation of the simulation model.

Response of the authors:

We agree with the comment of the reviewer. Therefore we added the reference pertaining to the previous validation of the model: "Lahr M, van der Zee D, Luijckx G, et al. A simulation based approach for improving utilization of thrombolysis in acute brain infarction. *Medical Care* 2013;51(12):1101-5." [Methods, page 6, line 8].

Comment 6: Methods: continuing on the previous theme, and in addition to any references provided in response to my previous comment, the input parameters, the distribution shape and values as used in the baseline model and the scenarios, as well as their sources should be included in one (or more) comprehensive tables. Table 1 is potentially a starting point, but this refers to the data collected prospectively in the four centres, rather than what went into the simulation model.

Response of the authors:

We thank the reviewer for this valuable comment. In response we added two tables to the Supplementary file detailing the items mentioned by the reviewer [Tables S1, S2, Supplementary file_rebuttal]. Please note that the far right column of Table S1 describes the modifications made to the baseline model, per scenario. Sources, i.e., references underlying our choice of distributions are mentioned in Table 2 (Overview of scenarios) in the Main text. To facilitate a clear identification of sources, best practices are explicitly linked to scenarios, see Table 2 (middle column).

Comment 7: Figure 1: is this a high-level depiction of the pathway or the actual process flow implemented as part of the simulation model? In any case, as part of describing the simulation model we need to see the specific process map that acted as a blueprint for the model to be developed.

Response of the authors:

We agree with the comment of the reviewer. Therefore we added a specific process map describing the acute stroke pathway in more detail, and included this in the Supplementary file [Figure S1, Supplementary file_rebuttal].

Comment 8: Table 2: the under-reporting of the simulation model and its constituent elements makes it rather difficult to fully understand this part of the study. That said, it appears that many 'delays' were used as input rather than output parameters from the simulation (e.g. for validation purposes). In reality, 'delays' are typically the result of the mismatch between demand and supply, or of the way a care process is organised (or both). If feeding delays in to the model as an input parameter, then how would this parameter change if for example, we were to change the way the care process is organised or vary arrivals rates etc.? Isn't the whole idea of modelling to be able to assess the impact of changes on delays/waiting times (as well as other output parameters?) How valid is the model, if the delay/waiting time distributions are exogenous rather than generated by the model itself?

Response of the authors:

We thank the reviewer for raising this important point.

Table 2 included hypothetical interventions that we based on the observed intervals (in fact time distributions) depicted in Table 1. As such, the baseline model was deliberately changed based on input values obtained from either the literature or expert opinion. No changes were made in the set-up of the pathway, only the statistical distributions underlying time intervals and diagnostic steps in the pathway were modified to reflect scenarios studied. An overview of model distributions, per scenario, are

provided in the far right column of Table S1 in the Supplementary file [Table S1, Supplementary file_rebuttal].

Our definition of patient delays (and activity durations) includes both service and possible waiting times. This definition could indeed be criticized from an operations management perspective. After all, many organisational measures meant to expedite patient flow seek to reduce process times, i.e., not a waiting time per se. In some settings patient waiting times might be acknowledged as an output.

In our setting, i.e., the acute stroke pathway, actual waiting times are less of an issue in expediting patient flow. Acute stroke patients are low in number, while their urgencies are perceived as very high. Their high urgency usually allows them to queue jump, effectively reducing waiting times for intra-hospital services close to zero. On the other hand, their low number (in our case: around 2% of all emergency medical services transports; around two visits to the hospital emergency department on a daily basis) suggests that their impact on waiting times – in acquiring resources shared with a much larger other group of patients – is negligible. Also see recent work of Monks et al. 2017 (reference 4 in the Supplementary file) on this subject (pp. 62, Section 4.3.4).

In addition, starting from the above observations we decided to refrain from modelling “the competition for resources” as is common in discrete event simulation models, and develop a monte carlo simulation model instead.

Our choice of model and the definition of activity durations is further explained in the Supplementary file [Supplementary file_rebuttal], and in Table S1 of the Supplementary file [Table S1, Supplementary file_rebuttal].

Comment 9: Figure 2. The chart is not very clear and this is not only because of how small it is on paper. In any case, I am reserving judgement on the results section of the paper until my comments above are addressed/rebutted satisfactorily.

Response of the authors:

We agree with the comment of the reviewer. We decided to leave it out, as we feel it does not make a significant contribution to the manuscript, that would go beyond results already presented in Table 3.

Dalius Jatuzis (Reviewer 2): Thank you for possibility to review your manuscript. It is really very important research for optimising the delivery of acute stroke care.

Comment 1: I would suggest briefly describing the statistical methods used in the work (even if they were fairly simple in this work).

Response of the authors:

We agree with the comments of the reviewer. Therefore we added a sentence to the Methods section. It now reads: “Chi-square tests were used to compare categorical variables” [Methods, page 9, line 15-16]. We also added the results of the statistical analyses to Table 3 for IVT rate [Table 3. Main text].

Comment 2: The introduction could be slightly shortened, focusing on the main idea and task of the work.

Response of the authors:

We thank the reviewer for the valuable comments. In response, we shortened or removed sentences in the Introduction section [Introduction, page 3, line 8-9; 13; 21-22, page 4, line 4-5; 11-12; 18-19].

Comment 3: Did you not need the ethics approval for this work, in your opinion? If so, I suggest you mention it in the text.

Response of the authors:

We thank the reviewer for raising this important point. Therefore we added a sentence to the informed consent paragraph in the Methods section. It now reads: “As such no additional approval from our local ethics committee was required” [Methods, page 9, line 20-21].

FORMATTING AMENDMENTS (if any)

Required amendments will be listed here; please include these changes in your revised version:

- Statements

Please embed the following statement to your main document just before your reference list.

- a. Data sharing statement

Response of the authors:

We thank the editor for pointing out this omission. We have included a sentence to the manuscript. It now reads: “Data sharing statement: The data used and analysed in the current study is available from the corresponding author upon request” [Methods, Main text, page 16, line 1-3].

VERSION 2 – REVIEW

REVIEWER	Christos Vasilakis University of Bath, UK
-----------------	--

REVIEW RETURNED	10-Nov-2019
-------------

GENERAL COMMENTS	I thank the authors for taking the time and investing the effort to look into and address, one way or another, all of my comments. I believe this is a much improved manuscript and the new supplementary files add a lot of value to the reporting of the study.  In response to Comment 2, the authors mention in their rebuttal “We agree with the reviewer that the data we used as input might be perceived as seemingly old. However, the current organization of care, i.e., the flow of patients in the system that was studied has not changed. It is the same centralized organisational model for acute stroke care. No changes in the set-up of the pathway for acute stroke patients receiving intravenous thrombolysis have been implemented over the years. As such the context has remained similar.” The response is satisfactory and some text along the lines of these points should be added to the manuscript. The second strength of the study (page 4) reads “The generic modelling approach adopted could be extended to include patient data from other stroke care systems”. However, the fact the simulation model has been implemented using proprietary software (Plant Simulation) coupled by the fact the code is not made available makes it difficult to see how the paper supports this particular assertion. A summary of the limitations arising from the modelling assumptions, such as no capacity constraints, should feature more prominently on this bulleted list. The lack of capacity constrains in the model should be mentioned in the limitations paragraph in the Discussion section. There is no information in relation to the input parameter(s) that control stroke onset in the model. What is the probability distribution associated with this aspect of the model? Minor comments  Table S1, Choice of route, first responder etc.: in addition to frequency please include percentages where possible. Table S1, replace ‘triangle’ distribution by ‘triangular’. Page 8, line 9: ‘... entire stroke chain’ is not a very accurate description as the scope of your study addresses the acute phase of stroke care (and not hospitalisation, rehabilitation, hospital discharge, step down care etc.) Table 1: “Median time from hospital arrival to neurological examination, min (IQR): 2 (0-0)”. Check and correct IQR values. Table 2: The readability of the paper would be improved if Table 2 and Table S2 were merged into one single table in the main paper. I suggest you keep the basic format of S2, adding citation and other info as needed.
---

REVIEWER	Dalius Jatuzis Vilnius University Faculty of Medicine, Lithuania
REVIEW RETURNED	10-Nov-2019

GENERAL COMMENTS	Thank you for answers and corrections.
--

VERSION 2 – AUTHOR RESPONSE

Reviewer(s)' Comments to Author:

Reviewer: 2

Reviewer Name: Dalius Jatuzis

Institution and Country: Vilnius University Faculty of Medicine, Lithuania Please state any competing interests or state 'None declared': None declared

Please leave your comments for the authors below Thank you for answers and corrections.

Response of the authors:

We thank the reviewer for his positive response.

Reviewer: 1

Reviewer Name: Christos Vasilakis

Institution and Country: University of Bath, UK Please state any competing interests or state 'None declared': None declared

Please leave your comments for the authors below I thank the authors for taking the time and investing the effort to look into and address, one way or another, all of my comments. I believe this is a much improved manuscript and the new supplementary files add a lot of value to the reporting of the study.

Comment 1: In response to Comment 2, the authors mention in their rebuttal “We agree with the reviewer that the data we used as input might be perceived as seemingly old. However, the current organization of care, i.e., the flow of patients in the system that was studied has not changed. It is the same centralized organisational model for acute stroke care. No changes in the set-up of the pathway for acute stroke patients receiving intravenous thrombolysis have been implemented over the years. As such the context has remained similar.” The response is satisfactory and some text along the lines of these points should be added to the manuscript.

Response of the authors:

We thank the reviewer for his positive response, and agree that additional text should be added to the manuscript. It now reads: “In addition, pathway set-up of acute stroke patients receiving IVT remained similar over the years” [Discussion, page 23, line 7-8].

Comment 2: The second strength of the study (page 4) reads “The generic modelling approach adopted could be extended to include patient data from other stroke care systems”. However, the fact the simulation model has been implemented using proprietary software (Plant Simulation) coupled by the fact the code is not made available makes it difficult to see how the paper supports this particular assertion. A summary of the limitations arising from the modelling assumptions, such as no capacity constraints, should feature more prominently on this bulleted list.

Response of the authors:

We agree with the comment of the reviewer and are grateful for raising this important point. Therefore we changed the bullet list of limitations. It now reads: “A simulation-based approach, as presented in this paper, can be instrumental in facilitating a broad overview of the set-up and performance of stroke pathways”. “Effects of capacity constraints on patients’ waiting for care services are not explicitly modelled given their high priorities, allowing them to queue jump.” [Strengths and limitations, page 4, line 4-9].

Comment 3: The lack of capacity constrains in the model should be mentioned in the limitations paragraph in the Discussion section.

Response of the authors:

We agree with the comment of the reviewer. Therefore we added a sentence to the Discussion section. It now reads: “Finally, model assumptions excluded the possibility for capacity constraints influencing patient waiting times, as patients and/or transport queuing seldom occurs due to the high prioritization

that potential acute stroke patients receive throughout the pathway. However, we acknowledge that such constraints might occur in other stroke care systems” [Discussion, page 23, line 8-12].

Comment 4: There is no information in relation to the input parameter(s) that control stroke onset in the model. What is the probability distribution associated with this aspect of the model?

Response of the authors:

We thank the reviewer for this valuable comment. Therefore we decided to add a sentence to the Supplementary file. It now reads: “Timing of stroke onset on the day did not affect our model assumptions, because we did not expect any impact on capacity constraints. Out of office hours were not included because the hospital under study (University Medical Center Groningen) has 24/7/365 occupation of personnel and facilities for acute stroke treatment” [Supplementary file, Model – process map and data].

Minor comments

Comment 5: Table S1, Choice of route, first responder etc.: in addition to frequency please include percentages where possible.

Response of the authors:

We agree with the reviewer and adapted Table S1 accordingly [Table S1, Supplementary file].

Comment 6: Table S1, replace ‘triangle’ distribution by ‘triangular’.

Response of the authors:

We changed the text accordingly. It now reads: “triangular” [Table S1, Supplementary file].

Comment 7: Page 8, line 9: ‘... entire stroke chain’ is not a very accurate description as the scope of your study addresses the acute phase of stroke care (and not hospitalisation, rehabilitation, hospital discharge, step down care etc.)

Response of the authors:

We agree with the reviewer and added a sentence to the Methods section. It now reads: “Stroke care pathways can be described in several distinct phases: hyperacute (emergency), acute and rehabilitation. Within this study we focused on the hyperacute phase, ranging from symptom onset until acute treatment with IVT” [Methods, page 8, line 11-13].

Comment 8: Table 1: “Median time from hospital arrival to neurological examination, min (IQR): 2 (0-0)”. Check and correct IQR values.

Response of the authors:

We agree with the reviewer and are thankful for pointing this out. Subsequently we used the range instead of the IQR for this variable in Table 1 [Table 1, page 7].

Comment 9: Table 2: The readability of the paper would be improved if Table 2 and Table S2 were merged into one single table in the main paper. I suggest you keep the basic format of S2, adding citation and other info as needed.

Response of the authors:

We agree with the reviewer. In response, we merged tables 2 and S2 into one table (Table 2, main manuscript) [Table 2, page 11-12]. As a result Table S2 is removed from the Supplementary file. Text of both the main paper and the supplementary file have been adapted to reflect changes [Main text file, page 10, line 2; Main text file, page 13, line 17-18; Supplementary file, Input parameters].

VERSION 3 – REVIEW

REVIEWER	Christos Vasilakis
-----------------	--------------------

	University of Bath, UK
REVIEW RETURNED	04-Dec-2019

GENERAL COMMENTS	I am happy to accept in its current version and I hope the authors think the paper was improved as a result of the 2 revisions.
---